# VL-BERT: Pre-training of Generic Visual-Linguistic Representations

**Weijie Su**[1,2*], **Xizhou Zhu**[1,2*], **Yue Cao**[2], **Bin Li**[1], **Lewei Lu**[2], **Furu Wei**[2], **Jifeng Dai**[2†]
[1]University of Science and Technology of China
[2]Microsoft Research Asia
{jackroos,ezra0408}@mail.ustc.edu.cn,binli@ustc.edu.cn
{yuecao,lewlu,fuwei,jifdai}@microsoft.com

## Abstract

We introduce a new pre-trainable generic representation for visual-linguistic tasks, called Visual-Linguistic BERT (VL-BERT for short). VL-BERT adopts the simple yet powerful Transformer model as the backbone, and extends it to take both visual and linguistic embedded features as input. In it, each element of the input is either of a word from the input sentence, or a region-of-interest (RoI) from the input image. It is designed to fit for most of the visual-linguistic downstream tasks. To better exploit the generic representation, we pre-train VL-BERT on the massive-scale Conceptual Captions dataset, together with text-only corpus. Extensive empirical analysis demonstrates that the pre-training procedure can better align the visual-linguistic clues and benefit the downstream tasks, such as visual commonsense reasoning, visual question answering and referring expression comprehension. It is worth noting that VL-BERT achieved the first place of single model on the leaderboard of the VCR benchmark. Code is released at
https://github.com/jackroos/VL-BERT.

## 1 Introduction

Pre-training of generic feature representations applicable to a variety of tasks in a domain is a hallmark of the success of deep networks. Firstly in computer vision, backbone networks designed for and pre-trained on ImageNet (Deng et al., 2009) classification are found to be effective for improving numerous image recognition tasks. Recently in natural language processing (NLP), Transformer networks (Vaswani et al., 2017) pre-trained with "masked language model" (MLM) objective (Devlin et al., 2018) on large language corpus excel at a variety of NLP tasks.

Meanwhile, for tasks at the intersection of vision and language, such as image captioning (Young et al., 2014; Chen et al., 2015; Sharma et al., 2018), visual question answering (VQA) (Antol et al., 2015; Johnson et al., 2017; Goyal et al., 2017; Hudson & Manning, 2019), visual commonsense reasoning (VCR) (Zellers et al., 2019; Gao et al., 2019), there lacks such pre-trained generic feature representations. The previous practice is to combine base networks pre-trained for image recognition and NLP respectively in a task-specific way. The task-specific model is directly finetuned for the specific target task, without any generic visual-linguistic pre-training. The task-specific model may well suffer from overfitting when the data for the target task is scarce. Also, due to the task-specific model design, it is difficult to benefit from pre-training, where the pre-training task may well be different from the target. There lacks a common ground for studying the feature design and pre-training of visual-linguistic tasks in general.

In the various network architectures designed for different visual-linguistic tasks, a key goal is to effectively aggregate the multi-modal information in both the visual and linguistic domains. For example, to pick the right answer in the VQA task, the network should empower integrating linguistic information from the question and the answers, and aggregating visual information from the input image, together with aligning the linguistic meanings with the visual clues. Thus, we seek to derive generic representations that can effectively aggregate and align visual and linguistic information.

---

*Equal contribution. This work is done when Weijie Su and Xizhou Zhu are interns at Microsoft Research Asia. †Corresponding author.

In the meantime, we see the successful application of Transformer attention (Vaswani et al., 2017) in NLP, together with its MLM-based pre-training technique in BERT (Devlin et al., 2018). The attention module is powerful and flexible in aggregating and aligning word embedded features in sentences, while the pre-training in BERT further enhances the capability.

Inspired by that, we developed VL-BERT, a pre-trainable generic representation for visual-linguistic tasks, as shown in Figure 1. The backbone of VL-BERT is of (multi-modal) Transformer attention module taking both visual and linguistic embedded features as input. In it, each element is either of a word from the input sentence, or a region-of-interest (RoI) from the input image, together with certain special elements to disambiguate different input formats. Each element can adaptively aggregate information from all the other elements according to the compatibility defined on their contents, positions, categories, and etc. The content features of a word / an RoI are domain specific (WordPiece embeddings (Wu et al., 2016) as word features, Fast R-CNN (Girshick, 2015) features for RoIs). By stacking multiple layers of multi-modal Transformer attention modules, the derived representation is of rich capability in aggregating and aligning visual-linguistic clues. And task-specific branches can be added above for specific visual-linguistic tasks.

To better exploit the generic representation, we pre-train VL-BERT at both large visual-linguistic corpus and text-only datasets[1]. The pre-training loss on the visual-linguistic corpus is incurred via predicting randomly masked words or RoIs. Such pre-training sharpens the capability of VL-BERT in aggregating and aligning visual-linguistic clues. While the loss on the text-only corpus is of the standard MLM loss in BERT, improving the generalization on long and complex sentences.

Comprehensive empirical evidence demonstrates that the proposed VL-BERT achieves state-of-the-art performance on various downstream visual-linguistic tasks, such as visual commonsense reasoning, visual question answering and referring expression comprehension. In particular, we achieved the first place of single model on the leaderboard of visual commonsense reasoning.

## 2    RELATED WORK

**Pre-training for Computer Vision** Prior to the era of deep networks, it is far from mature to share features among different tasks and to improve the features via pre-training. The models for various computer vision tasks are of too diverse design choices to derive a generic representation. With the success of AlexNet (Krizhevsky et al., 2012) in ImageNet (Deng et al., 2009) classification, we see the renaissance of convolutional neural networks (CNNs) in the vision community. Soon after that, researchers found that ImageNet pre-trained CNNs can serve well as generic feature representation for various downstream tasks (Donahue et al., 2014), such as object detection (Girshick et al., 2014), semantic segmentation (Long et al., 2015), instance segmentation (Hariharan et al., 2014). The improvement in backbone networks for ImageNet classification further improves the downstream tasks. Recently there are research works on directly training CNNs from scratch on massive-scale target datasets, without ImageNet pre-training (He et al., 2018). They achieved performance on par with those with ImageNet pre-training. While they also note that pre-training on a proper massive dataset is vital for improving performance on target tasks with scarce data.

**Pre-training for Natural Language Processing (NLP)** It is interesting to note that the development of pre-training techniques in NLP lags quite behind computer vision. There are previous research works on improving word embedding (Mikolov et al., 2013; Pennington et al., 2014; Kiros et al., 2015), which is a low-level linguistic feature representation. On top of that, numerous diverse architectures are designed for various NLP tasks. In the milestone work of Transformers (Vaswani et al., 2017), the Transformer attention module is proposed as a generic building block for various NLP tasks. After that, a serious of approaches are proposed for pre-training the generic representation, mainly based on Transformers, such as GPT (Radford et al., 2018), BERT (Devlin et al., 2018), GPT-2 (Radford et al., 2019), XLNet (Yang et al., 2019), XLM (Lample & Conneau, 2019), and RoBERTa (Liu et al., 2019). Among them, BERT is perhaps the most popular one due to its simplicity and superior performance.

**Pre-training for Visual-Linguistic Tasks.** The development course of models for visual-linguistic tasks is also quite similar to those in the computer vision and NLP communities. Previously, task-

---

[1]Here we exploit the Conceptual Captions dataset (Sharma et al., 2018) as the visual-linguistic corpus, and the BooksCorpus (Zhu et al., 2015) & English Wikipedia as the text-only corpus.

specific models are designed, wherein the features derived from off-the-shelf computer vision and NLP models are combined in an ad-hoc way for specific tasks. Model training is performed on the dataset for the specific task only.

VideoBERT (Sun et al., 2019b) is the first work seeking to conduct pre-training for visual-linguistic tasks. In it, video clips are processed by off-the-shelf networks for action recognition, and are assigned to different clusters (visual words) based on the derived features. The pre-training loss is incurred via predicting the cluster ids of masked video clips. Due to the abrupt clustering of the video clips, it losses considerable visual content information and hinders updating visual network parameters. In the following work of CBT (Sun et al., 2019a), such clustering mechanism is removed. Both works are applied on videos, which are of linear structure in the time dimension, same as sentences. It is highly desired to study at the well-established image-based visual-linguistic tasks.

Concurrent to our work, multiple works released on Arxiv very recently also seek to derive a pre-trainable generic representation for visual-linguistic tasks. Table 5 in Appendix compares among them. We briefly discuss some of these works here.

In ViLBERT (Lu et al., 2019) and LXMERT (Tan & Bansal, 2019), which are under review or just got accepted, the network architectures are of two single-modal networks applied on input sentences and images respectively, followed by a cross-modal Transformer combining information from the two sources. The attention pattern in the cross-modal Transformer is restricted, where the authors believe to improve the performance. The authors of ViLBERT claim that such two-stream design is superior than a single-stream unified model. Meanwhile, in the proposed VL-BERT, it is of a unified architecture based on Transformers without any restriction on the attention patterns. The visual and linguistic contents are fed as input to VL-BERT, wherein they interact early and freely. We found that our unified model of VL-BERT outperforms such two-stream designs.

VisualBert (Li et al., 2019b), B2T2 (Alberti et al., 2019), and Unicoder-VL (Li et al., 2019a), which are of work in progress or under review, are also of unified single-stream architecture. The differences of these works are compared in Table 5. The concurrent emergency of these research works indicates the importance of deriving a generic pre-trainable representation for visual-linguistic tasks.

In addition, there are three noticeable differences between VL-BERT and other concurrent works in pre-training. Their effects are validated in Section 4.3. (1) We found the task of Sentence-Image Relationship Prediction used in all of the other concurrent works (e.g., ViLBERT (Lu et al., 2019) and LXMERT (Tan & Bansal, 2019)) is of no help in pre-training visual-linguistic representations. Thus such a task is not incorporated in VL-BERT. (2) We pre-train VL-BERT on both visual-linguistic and text-only datasets. We found such joint pre-training improves the generalization over long and complex sentences. (3) Improved tuning of the visual representation. In VL-BERT, the parameters of Fast R-CNN, deriving the visual features, are also updated. To avoid visual clue leakage in the pre-training task of Masked RoI Classification with Linguistic Clues, the masking operation is conducted on the input raw pixels, other than the feature maps produced by layers of convolution.

## 3   VL-BERT

### 3.1   REVISIT BERT MODEL

Let $x = \{x_1, ..., x_N\}$ be the input elements in BERT (Devlin et al., 2018), which are of embedded features encoding sentence words. They are processed by a multi-layer bidirectional Transformer (Vaswani et al., 2017), where the embedding features of each element are transformed layer-by-layer in the fashion of aggregating features from the other elements with adaptive attention weights. Let $x^l = \{x_1^l, ..., x_N^l\}$ be the features of the $l$-th layer ($x^0$ is set as the input $x$). The features of the $(l+1)$-th layer, $x^{l+1}$, is computed by

$$\tilde{h}_i^{l+1} = \sum_{m=1}^{M} W_m^{l+1}\Big\{\sum_{j=1}^{N} A_{i,j}^m \cdot V_m^{l+1} x_j^l\Big\} \qquad \text{Multi-head Attention,} \qquad (1)$$

$$h_i^{l+1} = \text{LayerNorm}(x_i^l + \tilde{h}_i^{l+1}) \qquad \text{Residual Connection,} \qquad (2)$$

$$\tilde{x}_i^{l+1} = W_2^{l+1} \cdot \text{GELU}(W_1^{l+1} h_i^{l+1} + b_1^{l+1}) + b_2^{l+1} \qquad \text{Feed-forward,} \qquad (3)$$

$$x_i^{l+1} = \text{LayerNorm}(h_i^{l+1} + \tilde{x}_i^{l+1}) \qquad \text{Residual Connection,} \qquad (4)$$

where $m$ in Eq. 1 indexes over the attention heads, and $A_{i,j}^m \propto \exp[(Q_m^{l+1}x_i^l)^T(K_m^{l+1}x_j^l)]$ denotes the attention weights between elements $i$ and $j$ in the $m$-th head, which is normalized by $\sum_{j=1}^N A_{i,j}^m = 1$. $W_m^{l+1}$, $Q_m^{l+1}$, $K_m^{l+1}$ and $V_m^{l+1}$ are learnable weights for $m^{\text{th}}$ attention head, $W_1^{l+1}, W_2^{l+1}$ and $b_1^{l+1}, b_2^{l+1}$ in Eq. 3 are learnable weights and biases, respectively. Note that, the operations in Eq. $1 \sim 4$ is irrelevant to the order of input sequence, i.e. the final BERT representation of permuted input is same as the final BERT representation of the original input after the same permutation. The position of an element in BERT is encoded in its own embedding features by sequence positional embedding. Thanks to such decoupled representation, the BERT model is flexible enough to be pre-trained and finetuned for a variety of NLP tasks.

In BERT pre-training, the masked language modeling (MLM) task is introduced. The embedded features of a certain input word would be randomly masked out (the token embedding channels capturing the word content is replaced by a special [MASK] token). The BERT model is trained to predict the masked word from linguistic clues of all the other unmasked elements. As explained in Wang & Cho (2019), the overall MLM-based training of BERT is equivalent to optimizing the following joint probability distribution

$$\log P(x|\theta) = \frac{1}{Z(\theta)} \sum_{i=1}^N \log \phi_i(x|\theta), \tag{5}$$

where $\phi_i(x|\theta)$ is the potential function for the $i$-th input element, with parameters $\theta$, and $Z(\theta)$ is the partition function. Each log-potential term $\log \phi_i(x)$ is defined as

$$\log \phi_i(x|\theta) = x_i^T f_i(x_{\backslash i}|\theta)_i, \tag{6}$$

where $f_i(x_{\backslash i}|\theta)$ denotes the final output feature of BERT corresponding to the $i$-th element for input $x_{\backslash i}$, where $x_{\backslash i}$ is defined as $x_{\backslash i} = \{x_1, ..., x_{i-1}, [\text{MASK}], x_{i+1}, ..., x_N\}$. The incurred MLM-based loss is as

$$L_{\text{MLM}}(\theta) = -E_{x \sim D, i \sim \{1,...,N\}} \log \phi_i(x), \tag{7}$$

where $x$ is a randomly sampled sentence from the training set $D$, and $i$ is a randomly sampled location for masking words.

The second pre-training task, Next Sentence Prediction, focuses on modeling the relationship between two sentences. Two sentences are sampled from the input document, and the model should predict whether the second sentence is the direct successor of the first. In BERT, the sampled two sentences are concatenated into one input sequence, with special elements [CLS] and [SEP] inserted prior to the first and the second sentences, respectively. A Sigmoid classifier is appended on the final output feature corresponding to the [CLS] element to make the prediction. Let $x$ be the input sequence, $t \in \{0, 1\}$ indicates the relationship between the two sentences. The loss function is defined as

$$L_{\text{NSP}}(\theta) = -E_{(x,t) \sim D}\left[t \log(g(x_0^L)) + (1-t)\log(1 - g(x_0^L))\right], \tag{8}$$

where $x_0^L$ is the final output feature of the [CLS] element (at the $L$-th layer), and $g(x_0^L)$ is the classifier output.

## 3.2 MODEL ARCHITECTURE

Figure 1 illustrates the architecture of VL-BERT. Basically, it modifies the original BERT (Devlin et al., 2018) model by adding new elements to accommodate the visual contents, and a new type of visual feature embedding to the input feature embeddings. Similar to BERT, the backbone is of multi-layer bidirectional Transformer encoder (Vaswani et al., 2017), enabling dependency modeling among all the input elements. Different to BERT processing sentence words only, VL-BERT takes both visual and linguistic elements as input, which are of features defined on regions-of-interest (RoIs) in images and sub-words from input sentences, respectively. The RoIs can either be bounding boxes produced by object detectors, or be annotated ones in certain tasks.

It is worth noting that the input formats vary for different visual-linguistic tasks (e.g., <Caption, Image> for image captioning, and <Question, Answer, Image> for VQA (Antol et al., 2015; Johnson et al., 2017; Goyal et al., 2017; Hudson & Manning, 2019) and VCR (Zellers et al., 2019; Gao et al., 2019)). But thanks to the unordered representation nature of Transformer attention (e.g., the

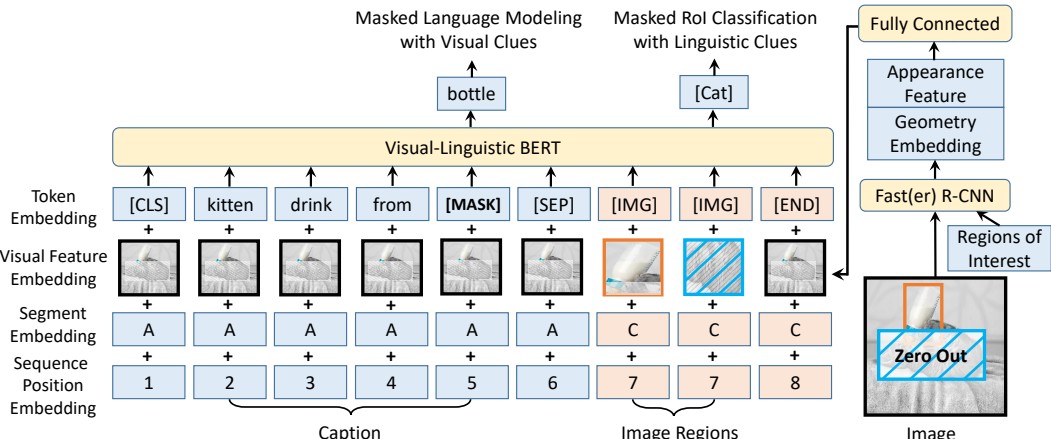

Figure 1: Architecture for pre-training VL-BERT. All the parameters in this architecture including VL-BERT and Fast R-CNN are jointly trained in both pre-training and fine-tuning phases.

position of a word in sentence is encoded by the positional embedding only, other than the order in the input sequence), a generic representation can be derived as long as the input elements and embedding features are properly designed. Three types of input elements are involved, namely, visual, linguistic, and special elements for disambiguating different input formats. The input sequence always starts with a special classification element ([CLS]), then goes on with linguistic elements, then follows up with visual elements, and ends with a special ending element ([END]). A special separation element ([SEP]) is inserted in between different sentences in the linguistic elements, and between the linguistic and visual elements. For each input element, its embedding feature is the summation of four types of embedding, namely, token embedding, visual feature embedding, segment embedding, and sequence position embedding. Among them, the visual feature embedding is newly introduced for capturing visual clues, while the other three embeddings follow the design in the original BERT paper.

**Token Embedding** Following the practice in BERT, the linguistic words are embedded with Word-Piece embeddings (Wu et al., 2016) with a 30,000 vocabulary. A special token is assigned to each special element. For the visual elements, a special [IMG] token is assigned for each one of them.

**Visual Feature Embedding** We firstly describe visual appearance feature and visual geometry embedding separately, and then how to combine them to form the visual feature embedding.

For the visual element corresponding to an RoI, the visual appearance feature is extracted by applying a Fast R-CNN (Girshick, 2015) detector (i.e., the detection branch in Faster R-CNN (Ren et al., 2015)), where the feature vector prior to the output layer of each RoI is utilized as the visual feature embedding (of 2048-d in paper). For the non-visual elements, the corresponding visual appearance features are of features extracted on the whole input image. They are obtained by applying Faster R-CNN on an RoI covering the whole input image.

The visual geometry embedding is designed to inform VL-BERT the geometry location of each input visual element in image. Each RoI is characterized by a 4-d vector, as $\left(\frac{x_{\text{LT}}}{W}, \frac{y_{\text{LT}}}{H}, \frac{x_{\text{RB}}}{W}, \frac{h_{\text{RB}}}{H}\right)$, where $(x_{\text{LT}}, y_{\text{LT}})$ and $(x_{\text{RB}}, y_{\text{RB}})$ denote the coordinate of the top-left and bottom-right corner respectively, and $W, H$ are of the width and height of the input image. Following the practice in Relation Networks (Hu et al., 2018), the 4-d vector is embedded into a high-dimensional representation (of 2048-d in paper) by computing sine and cosine functions of different wavelengths.

The visual feature embedding is attached to each of the input elements, which is the output of a fully connected layer taking the concatenation of visual appearance feature and visual geometry embedding as input.

**Segment Embedding** Three types of segment, $A, B, C$, are defined to separate input elements from different sources, namely, $A$ and $B$ for the words from the first and second input sentence respectively, and $C$ for the RoIs from the input image. For example, for input format of <Question, Answer, Image>, $A$ denotes Question, $B$ denotes Answer, and $C$ denotes Image. For input format

of <Caption, Image>, $A$ denotes Caption, and $C$ denotes Image. A learned segment embedding is added to every input element for indicating which segment it belongs to.

**Sequence Position Embedding** A learnable sequence position embedding is added to every input element indicating its order in the input sequence, same as BERT. Because there is no natural order among input visual elements, any permutation of them in the input sequence should achieve the same result. Thus the sequence position embedding for all visual elements are the same.

### 3.3 PRE-TRAINING VL-BERT

The generic feature representation of VL-BERT enables us to pre-train it on massive-scale datasets, with properly designed pre-training tasks. We pre-train VL-BERT on both visual-linguistic and text-only datasets. Here we utilize the Conceptual Captions dataset (Sharma et al., 2018) as the visual-linguistic corpus. It contains around 3.3 million images annotated with captions, which are harvested from web data and processed through an automatic pipeline. The issue with the Conceptual Captions dataset is that the captions are mainly simple clauses, which are too short and simple for many down-stream tasks. To avoid overfitting on such short and simple text scenario, we also pre-train VL-BERT on text-only corpus with long and complex sentences. We utilize the BooksCorpus (Zhu et al., 2015) and the English Wikipedia datasets, which are also utilized in pre-training BERT.

In SGD training, in each mini-batch, samples are randomly drawn from both Conceptual Captions and BooksCorpus & English Wikipedia (at a ratio of 1:1). For a sample drawn from Conceptual Captions, the input format to VL-BERT is of <Caption, Image>, where the RoIs in the image are localized and categorized by a pre-trained Faster R-CNN object detector. Two pre-training tasks are exploited to incur loss, which are as follows.

*Task #1*: *Masked Language Modeling with Visual Clues* This task is very similar to the Masked Language Modeling (MLM) task utilized in BERT. The key difference is that visual clues are incorporated in VL-BERT for capturing the dependencies among visual and linguistic contents. During pre-training, each word in the input sentence(s) is randomly masked (at a probability of 15%). For the masked word, its token is replaced with a special token of [MASK]. The model is trained to predict the masked words, based on the unmasked words and the visual features. The task drives the network to not only model the dependencies in sentence words, but also to align the visual and linguistic contents. For example, in Figure 1 "kitten drinking from [MASK]", without the input image, the masked word could be any containers, such as "bowl", "spoon" and "bottle". The representation should capture the correspondence of the word "bottle" and the corresponding RoIs in the image to make the right guess. During pre-training, the final output feature corresponding to the masked word is fed into a classifier over the whole vocabulary, driven by Softmax cross-entropy loss.

*Task #2*: *Masked RoI Classification with Linguistic Clues* This is a dual task of Task #1. Each RoI in image is randomly masked out (with 15% probability), and the pre-training task is to predict the category label of the masked RoI from the other clues. To avoid any visual clue leakage from the visual feature embedding of other elements, the pixels laid in the masked RoI are set as zeros before applying Fast R-CNN. During pre-training, the final output feature corresponding to the masked RoI is fed into a classifier with Softmax cross-entropy loss for object category classification. The category label predicted by pre-trained Faster R-CNN is set as the ground-truth. An example is shown in Figure 1. The RoI corresponding to cat in image is masked out, and the corresponding category cannot be predicted from any visual clues. But with the input caption of "kitten drinking from bottle", the model can infer the category by exploiting the linguistic clues.

For a sample drawn from the BooksCorpus & English Wikipedia datasets, the input format to VL-BERT degenerates to be <Text, $\varnothing$ >, where no visual information is involved. The "visual feature embedding" term in Figure 1 is a learnable embedding shared for all words. The training loss is from the standard task of Masked Language Modeling (MLM) as in BERT.

In summary, the pre-training on visual-linguistic corpus improves the detailed alignment between visual and linguistic contents. Such detailed alignment is vital for many downstream tasks (for example, in Visual Grounding (Kazemzadeh et al., 2014), the model locates the most relevant object or region in an image based on a natural language query). While the pre-training on text-only corpus facilitates downstream tasks involving understanding of long and complex sentences.

### 3.4 FINE-TUNING VL-BERT

VL-BERT is designed to be a generic feature representation for various visual-linguistic tasks. It is relatively simple to finetune VL-BERT for various downstream tasks. We simply need to feed VL-BERT with properly formatted input and output, and finetune all the network parameters end-to-end. For the input, the typical formats of <Caption, Image> and <Question, Answer, Image> cover the majority visual-linguistic tasks. VL-BERT also supports more sentences and more images as long as appropriate segment embeddings are introduced to identify different input sources. At the output, typically, the final output feature of the [CLS] element is used for sentence-image-relation level prediction. The final output features of words or RoIs are for word-level or RoI-level prediction. In addition to the input and output format, task-specific loss functions and training strategies also need to be tuned. See Section 4.2 for the detailed design choices and settings.

## 4 EXPERIMENT

### 4.1 PRE-TRAINING

As described in Section 3.3, we pre-train VL-BERT jointly on Conceptual Captions (Sharma et al., 2018) as visual-linguistic corpus, and BooksCorpus (Zhu et al., 2015) & English Wikipedia as text-only corpus. As VL-BERT is developed via adding new inputs capturing visual information to the original BERT model, we initialize the parameters to be the same as the original BERT described in (Devlin et al., 2018). VL-BERT$_{BASE}$ and VL-BERT$_{LARGE}$ denote models developed from the original BERT$_{BASE}$ and BERT$_{LARGE}$ models, respectively. The newly added parameters in VL-BERT are randomly initialized from a Gaussian distribution with mean of 0 and standard deviation of 0.02. Visual content embedding is produced by Faster R-CNN + ResNet-101, initialized from parameters pre-trained on Visual Genome (Krishna et al., 2017) for object detection (see BUTD (Anderson et al., 2018)).

Prior to pre-training on Conceptual Captions, the pre-trained Faster R-CNN is applied to extract RoIs. Specifically, at most 100 RoIs with detection scores higher than 0.5 are selected for each image. At minimum, 10 RoIs are selected from one image, regardless of the detection score threshold. The detailed parameter settings are in Appendix.

### 4.2 FINE-TUNING ON DOWNSTREAM TASKS

The pre-trained VL-BERT model can be fine-tuned for various downstream visual-linguistic tasks, with simple modifications on the input format, output prediction, loss function and training strategy.

#### 4.2.1 VISUAL COMMONSENSE REASONING (VCR)

| Model | Q → A | | QA → R | | Q → AR | |
|---|---|---|---|---|---|---|
| | val | test | val | test | val | test |
| R2C (Zellers et al., 2019) | 63.8 | 65.1 | 67.2 | 67.3 | 43.1 | 44.0 |
| ViLBERT (Lu et al., 2019)[†] | 72.4 | 73.3 | 74.5 | 74.6 | 54.0 | 54.8 |
| VisualBERT (Li et al., 2019b)[†] | 70.8 | 71.6 | 73.2 | 73.2 | 52.2 | 52.4 |
| B2T2 (Alberti et al., 2019)[†] | 71.9 | 72.6 | 76.0 | 75.7 | 54.9 | 55.0 |
| VL-BERT$_{BASE}$ w/o pre-training | 73.1 | - | 73.8 | - | 54.2 | - |
| VL-BERT$_{BASE}$ | 73.8 | - | 74.4 | - | 55.2 | - |
| VL-BERT$_{LARGE}$ | 75.5 | 75.8 | 77.9 | 78.4 | 58.9 | 59.7 |

Table 1: Comparison to the state-of-the-art methods with single model on the VCR dataset.
† indicates concurrent works.

Visual Commonsense Reasoning (VCR) focuses on higher-order cognitive and commonsense understanding of the given image. In the dataset of Zellers et al. (2019), given an image and a list of categorized RoIs, a question at cognition level is raised. The model should pick the right answer to the question and provide the rationale explanation. For each question, there are 4 candidate answers and 4 candidate rationales. This holistic task (Q → AR) is decomposed into two sub-tasks wherein researchers can train specific individual models: question answering (Q → A) and answer

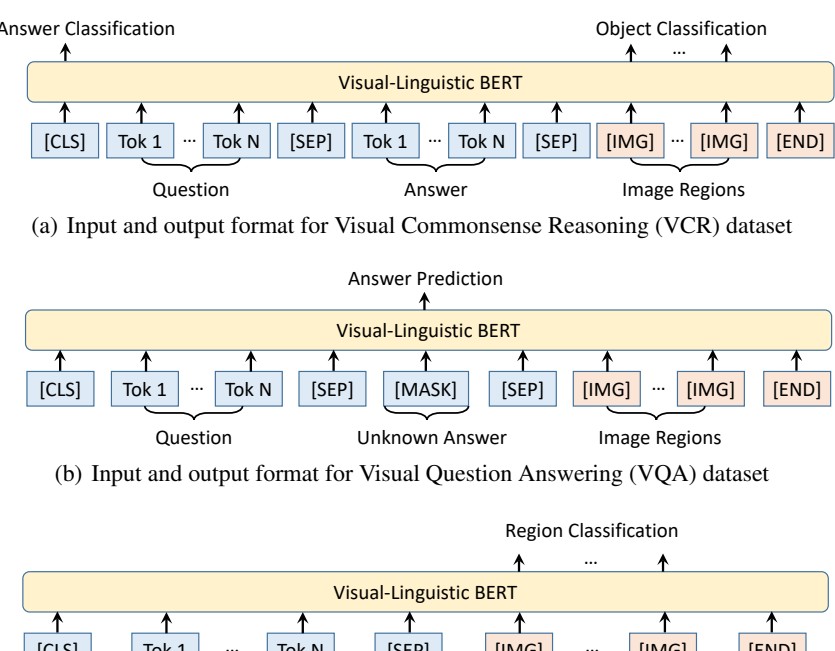

(a) Input and output format for Visual Commonsense Reasoning (VCR) dataset

(b) Input and output format for Visual Question Answering (VQA) dataset

(c) Input and output format for Referring Expression task on RefCOCO+ dataset

Figure 2: Input and output formats for fine-tuning different visual-linguistic downstream tasks.

justification (QA → R). The released VCR dataset consists of 265k pairs of questions, answers, and rationales, over 100k unique movie scenes (100k images). They are split into training, validation, and test sets consisting of 213k questions and 80k images, 27k questions and 10k images, and 25k questions and 10k images, respectively.

Our experimental protocol for VCR follows that in R2C (Zellers et al., 2019). The model is trained on the train split, and is evaluated at the val and test sets. In the original work R2C, task-specific "Grounding", "Contextualization" and "Reasoning" modules are designed. Here we simply adopt the generic representation of VL-BERT for the task. Figure 2 (a) illustrates the input format, <Question, Answer, Image>. For the sub-task of Q → A, 'Q' and 'A' are filled to the Question section and Answer section respectively. For the sub-task of QA → R , the concatenation of 'Q' and 'A' is filled to the Question section, and 'R' is filled to the Answer section. The input RoIs to VL-BERT are the ground-truth annotations in the dataset. The final output feature of [CLS] element is fed to a Softmax classifier for predicting whether the given Answer is the correct choice. During fine-tuning, we adopt two losses, the classification over the correctness of the answers and the RoI classification with linguistic clues. The detailed parameter settings are in Appendix.

Table 1 presents the experiment results. Pre-training VL-BERT improves the performance by 1.0% in the final Q → AR task, which validates the effectiveness of pre-training. Compared with R2C, we do not use ad-hoc task-specific modules. Instead, we simply adopt the generic representation of VL-BERT and jointly train the whole model end-to-end. Despite the same input, output and experimental protocol as R2C, VL-BERT outperforms R2C by large margins, indicating the power of our simple cross-modal architecture. Compared with other concurrent works, i.e., ViLBERT, VisualBERT and B2T2, our VL-BERT achieves the state-of-the-art performance.

### 4.2.2 VISUAL QUESTION ANSWERING (VQA)

In the VQA task, given a natural image, a question at the perceptual level is asked, and the algorithm should generate / choose the correct answer. Here we conduct experiments on the widely-used VQA v2.0 dataset (Goyal et al., 2017), which is built based on the COCO (Lin et al., 2014) images. The VQA v2.0 dataset is split into train (83k images and 444k questions), validation (41k images and

| Model | test-dev | test-std |
|---|---|---|
| BUTD (Anderson et al., 2018) | 65.32 | 65.67 |
| ViLBERT (Lu et al., 2019)[†] | 70.55 | 70.92 |
| VisualBERT (Li et al., 2019b)[†] | 70.80 | 71.00 |
| LXMERT (Tan & Bansal, 2019)[†] | 72.42 | 72.54 |
| VL-BERT$_{BASE}$ w/o pre-training | 69.58 | - |
| VL-BERT$_{BASE}$ | 71.16 | - |
| VL-BERT$_{LARGE}$ | 71.79 | 72.22 |

Table 2: Comparison to the state-of-the-art methods with single model on the VQA dataset. † indicates concurrent works.

214k questions), and test (81k images and 448k questions) sets. Following the experimental protocol in BUTD (Anderson et al., 2018), for each question, the algorithm should pick the corresponding answer from a shared set consisting of 3,129 answers.

Figure 2 (b) illustrates the input format for the VQA task, which is of <Question, Answer, Image>. As the possible answers are from a shared pool independent to the question, we only fill a [MASK] element to the Answer section. As in BUTD (Anderson et al., 2018), the input RoIs in VL-BERT are generated by a Faster R-CNN detector pre-trained on Visual Genome (Krishna et al., 2017). The answer prediction is made from a multi-class classifier based upon the output feature of the [MASK] element. During fine-tuning, the network training is driven by the multi-class cross-entropy loss over the possible answers. The detailed parameter settings are in Appendix.

Table 2 presents our experimental results. Pre-training VL-BERT improves the performance by 1.6%, which validates the importance of pre-training. VL-BERT shares the same input (i.e., question, image, and RoIs), output and experimental protocol with BUTD, a prevalent model specifically designed for the task. Still, VL-BERT surpasses BUTD by over 5% in accuracy. Except for LXMERT, our VL-BERT achieves better performance than the other concurrent works. This is because LXMERT is pre-trained on massive visual question answering data (aggregating almost all the VQA datasets based on COCO and Visual Genome). While our model is only pre-trained on captioning and text-only dataset, where there is still gap with the VQA task.

### 4.2.3 REFERRING EXPRESSION COMPREHENSION

| Model | Ground-truth Regions | | | Detected Regions | | |
|---|---|---|---|---|---|---|
| | val | testA | testB | val | testA | testB |
| MAttNet (Yu et al., 2018) | 71.01 | 75.13 | 66.17 | 65.33 | 71.62 | 56.02 |
| ViLBERT (Lu et al., 2019)[†] | - | - | - | 72.34 | 78.52 | 62.61 |
| VL-BERT$_{BASE}$ w/o pre-training | 74.41 | 77.28 | 67.52 | 66.03 | 71.87 | 56.13 |
| VL-BERT$_{BASE}$ | 79.88 | 82.40 | 75.01 | 71.60 | 77.72 | 60.99 |
| VL-BERT$_{LARGE}$ | 80.31 | 83.62 | 75.45 | 72.59 | 78.57 | 62.30 |

Table 3: Comparison to the state-of-the-art methods with single model on the RefCOCO+ dataset. † indicates concurrent work.

A referring expression is a natural language phrase that refers to an object in an image. The referring expression comprehension task is to localize the object in an image with the given referring expression. We adopt the RefCOCO+ (Kazemzadeh et al., 2014) dataset for evaluation, consisting of 141k expressions for 50k referred objects in 20k images in the COCO dataset (Lin et al., 2014). The referring expressions in RefCOCO+ are forbidden from using absolute location words, e.g. left dog. Therefore the referring expressions focus on purely appearance-based descriptions. RefCOCO+ are split into four sets, training set (train), validation set (val), and two testing sets (testA and testB). Images containing multiple people are in testA set, while images containing multiple objects of other categories are in testB set. There is no overlap between the training, validation and testing images.

Figure 2 (c) illustrates the input format for referring expression comprehension , where the input format is of <Query, Image>. Model training and evaluation are conducted either on the ground-truth RoIs or on the detected boxes in MAttNet (Yu et al., 2018). And the results are reported either in the track of ground-truth regions or that of detected regions, respectively. During training, we

compute the classification scores for all the input RoIs. For each RoI, a binary classification loss is applied. During inference, we directly choose the RoI with the highest classification score as the referred object of the input referring expression. The detailed parameter settings are in Appendix.

Table 3 presents our experimental results. Pre-trained VL-BERT significantly improves the performance. Compared with MAttNet, VL-BERT is much simpler without task-specific architecture designs, yet much better. VL-BERT achieves comparable performance with the concurrent work of ViLBERT.

## 4.3 ABLATION STUDY

| Settings | Masked Language Modeling with Visual Clues | Masked RoI Classification with Linguistic Clues | Sentence-Image Relationship Prediction | with Text-only Corpus | Tuning Fast R-CNN | VCR | | VQA | RefCOCO+ Detected Regions |
|---|---|---|---|---|---|---|---|---|---|
| | | | | | | Q→A val | QA→R val | test-dev | val |
| w/o pre-training | | | | | | 72.9 | 73.0 | 69.5 | 62.7 |
| (a) | ✓ | | | | | 72.9 | 73.1 | 71.0 | 69.1 |
| (b) | ✓ | ✓ | | | | 73.0 | 73.1 | 71.1 | 70.7 |
| (c) | ✓ | ✓ | ✓ | | | 72.2 | 72.4 | 70.3 | 69.5 |
| (d) | ✓ | ✓ | | ✓ | | 73.4 | 73.8 | 71.1 | 70.7 |
| VL-BERT$_{BASE}$ | ✓ | ✓ | | ✓ | ✓ | 73.8 | 73.9 | 71.2 | 71.1 |

Table 4: Ablation study for VL-BERT$_{BASE}$ with $0.5\times$ fine-tuning epochs.

Table 4 ablates key design choices in pre-training VL-BERT. For experimental efficiency, the fine-tuning epoches of VL-BERT are of $0.5\times$ of those in Section 4.2, with only VL-BERT$_{BASE}$ model.

Overall, the pre-training of VL-BERT improves the performance over all the three down-stream tasks (by comparing setting "w/o pre-training" and VL-BERT$_{BASE}$). The improvement amplitude varies for different tasks. By comparing setting (a) to that of "w/o pre-training", we see the benefits of Task #1, Masked Language Modeling with Visual Clues. By further incorporating Task #2, Masked RoI Classification with Linguistic Clues, the accuracy further improves on RefCOCO+, but gets stuck at VCR and VQA. This might be because only RefCOCO+ utilizes the final output feature corresponding to [IMG] tokens for prediction. Thus the pre-training of such features is beneficial. Setting (c) incorporates the task of Sentence-Image Relationship Prediction as in ViLBERT (Lu et al., 2019) and LXMERT (Tan & Bansal, 2019). It would hurt accuracy on all the three down-stream tasks. We guess the reason is because the task of Sentence-Image Relationship Prediction would introduce unmatched image and caption pairs as negative examples. Such unmatched samples would hamper the training of other tasks. Setting (d) adds text-only corpus during pre-training. Compared with setting (b), it improves the performance over all three down-stream tasks, and is most significant on VCR. This is because the task of VCR involves more complex and longer sentences than those in VQA and RefCOCO+[2]. By further finetuning the network parameters of Fast R-CNN, which generates the visual features, we get the final setting of VL-BERT$_{BASE}$. Such end-to-end training of the entire network is helpful for all the downstream tasks.

## 5 CONCLUSION

In this paper, we developed VL-BERT, a new pre-trainable generic representation for visual-linguistic tasks. Instead of using ad-hoc task-specific modules, VL-BERT adopts the simple yet powerful Transformer model as the backbone. It is pre-trained on the massive-scale Conceptual Captions dataset, together with text-only corpus. Extensive empirical analysis demonstrates that the pre-training procedure can better align the visual-linguistic clues, and thus benefit the downstream tasks. In the future, we would like to seek better pre-training tasks, which could beneficial more downstream tasks (e.g., Image Caption Generation).

### ACKNOWLEDGMENTS

The work is partially supported by the National Natural Science Foundation of China under grand No.U19B2044 and No.61836011.

---

[2]In VCR, there are 16.0 and 33.7 words per sample on average for Q → A and QA → R sub-tasks, respectively. While the words per sample for VQA and RefCOCO+ are of 7.2 and 3.5, respectively.

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

# A Appendix

## A.1 Comparison among VL-BERT and other works

Table 5 compares among VL-BERT and other concurrent works for pre-training generic visual-linguistic representations.

| | Method | Architecture | Visual Token | Pre-train Datasets | Pre-train Tasks | Downstream Tasks |
|---|---|---|---|---|---|---|
| Published Works | VideoBERT (Sun et al., 2019b) | single cross-modal Transformer | video frame | Cooking312K (Sun et al., 2019b) | 1) sentence-image alignment 2) masked language modeling 3) masked visual-words prediction | 1) zero-shot action classification 2) video captioning |
| Works Under Review / Just Got Accepted | CBT (Sun et al., 2019a) | two single-modal Transformer (vision & language respectively) + one cross-modal Transformer | video frame | Cooking312K (Sun et al., 2019b) | 1) sentence-image alignment 2) masked language modeling 3) masked visual-feature regression | 1) action anticipation 2) video captioning |
| | ViLBERT (Lu et al., 2019) | one single-modal Transformer (language) + one cross-modal Transformer (with restricted attention pattern) | image RoI | Conceptual Captions (Sharma et al., 2018) | 1) sentence-image alignment 2) masked language modeling 3) masked visual-feature classification | 1) visual question answering 2) visual commonsense reasoning 3) grounding referring expressions 4) image retrieval 5) zero-shot image retrieval |
| | B2T2 (Alberti et al., 2019) | single cross-modal Transformer | image RoI | Conceptual Captions (Sharma et al., 2018) | 1) sentence-image alignment 2) masked language modeling | 1) visual commonsense reasoning |
| | LXMERT (Tan & Bansal, 2019) | two single-modal Transformer (vision & language respectively) + one cross-modal Transformer | image RoI | ‡ COCO Caption + VG Caption + VG QA + VQA + GQA | 1) sentence-image alignment 2) masked language modeling 3) masked visual-feature classification 4) masked visual-feature regression 5) visual question answering | 1) visual question answering 2) natural language visual reasoning |
| | VisualBERT (Li et al., 2019b) | single cross-modal Transformer | image RoI | COCO Caption (Chen et al., 2015) | 1) sentence-image alignment 2) masked language modeling | 1) visual question answering 2) visual commonsense reasoning 3) natural language visual reasoning 4) grounding phrases |
| | Unicoder-VL (Li et al., 2019a) | single cross-modal Transformer | image RoI | Conceptual Captions (Sharma et al., 2018) | 1) sentence-image alignment 2) masked language modeling 3) masked visual-feature classification | 1) image-text retrieval 2) zero-shot image-text retrieval |
| | Our VL-BERT | single cross-modal Transformer | image RoI | Conceptual Captions (Sharma et al., 2018) + BooksCorpus (Zhu et al., 2015) + English Wikipedia | 1) masked language modeling 2) masked visual-feature classification | 1) visual question answering 2) visual commonsense reasoning 3) grounding referring expressions |

‡ LXMERT is pre-trained on COCO Caption (Chen et al., 2015), VG Caption (Krishna et al., 2017), VG QA (Zhu et al., 2016), VQA (Antol et al., 2015) and GQA (Hudson & Manning, 2019).

Table 5: Comparison among our VL-BERT and other works seeking to derive pre-trainable generic representations for visual-linguistic tasks.

## A.2 Detailed experiment settings

Pre-training is conducted on 16 Tesla V100 GPUs for 250k iterations by SGD. In each mini-batch, 256 samples are drawn. Among them, 128 samples are of <Caption, Image> pairs from Conceptual Captions, and the rest 128 samples are sequential tokens (at most 64 tokens for each sequence) from BooksCorpus & English Wikipedia. In SGD, Adam optimizer (Kingma & Ba, 2014) is applied, with base learning rate of $2 \times 10^{-5}$, $\beta_1 = 0.9$, $\beta_2 = 0.999$, weight decay of $10^{-4}$, learning rate warmed up over the first 8,000 steps, and linear decay of the learning rate. All the parameters in VL-BERT and Fast R-CNN are jointly trained in both pre-training and fine-tuning phase. The visual feature input for textual corpus is a learnable embedding shared for all words. In the task of Masked RoI Classification with Linguistic Clues, the pixels lying in all the masked RoIs are set as zeros in the image. A box covering the whole image is added as a RoI and would not be masked.

For VCR, the fine-tuning is conducted on 16 Tesla V100 GPUs for 20 epochs. In each mini-batch, 256 triplets of <Question, Answer, Image> are sampled. In SGD, the basic mini-batch gradient descent is conducted, with base learning rate of $5 \times 10^{-3}$, momentum of 0.9, and weight decay of $10^{-4}$. The learning rate is linearly warmed up in the first 1,000 steps from an initial learning rate of 0, and is decayed by 0.1 at the 14-th and the 18-th epochs.

For VQA, the fine-tuning is conducted on 16 Tesla V100 GPUs for 20 epochs. In each mini-batch, 256 triplets of <Question, Answer, Image> are sampled. In SGD, Adam optimizer is applied, with base learning rate of $1 \times 10^{-4}$, $\beta_1 = 0.9$, $\beta_2 = 0.999$, weight decay of $10^{-4}$, learning rate warmed up over the first 2,000 steps, and linear decay of the learning rate.

For RefCOCO+, the fine-tuning is conducted on 16 Tesla V100 GPUs for 20 epochs. In each mini-batch, 256 pairs of <Query, Image> are sampled. In SGD, Adam optimizer is applied, with base learning rate of $1 \times 10^{-4}$, $\beta_1 = 0.9$, $\beta_2 = 0.999$, weight decay of $10^{-4}$, learning rate warmed up over the first 500 steps, and linear decay of the learning rate.

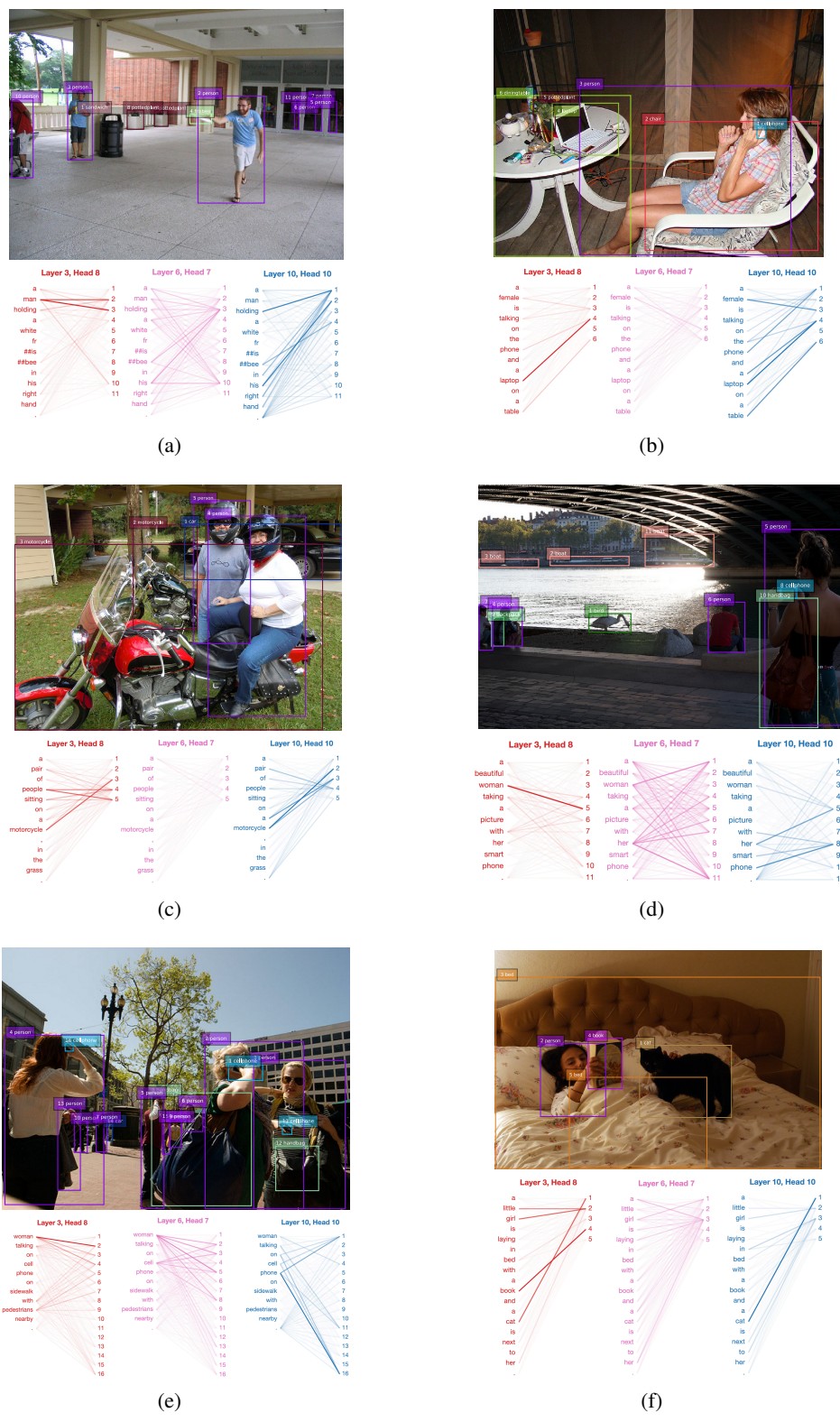

Figure 3: Visualization of attention maps in pre-trained VL-BERT_BASE. Line intensity indicates the magnitude of attention probability with the text token as query and the image RoI as key. The intensity is affinely rescaled to set the maximum value as 1 and the minimum as 0, across different heads in each layer. The index of network layer and attention head is counted from 0.

### A.3 VISUALIZATION OF ATTENTION MAPS IN VL-BERT

To better understand what VL-BERT learns from pre-training, we visualized the attention maps of pre-trained VL-BERT (without fine-tuning on downstream tasks) using BertViz[3](Vig, 2019).

Some visualization results on COCO (Lin et al., 2014; Chen et al., 2015) val2017 set are shown in Figure 3. We can see different attention patterns across attention heads. For some attention heads, text tokens attend more on the associated image RoIs. While in some other heads, text tokens attend uniformly to all RoIs. It demonstrates the ability of VL-BERT in aggregating and aligning visual-linguistic contents.

---

[3]`https://github.com/jessevig/bertviz`

