# OpenReview forum: "VL-BERT: Pre-training of Generic Visual-Linguistic Representations"
_ICLR.cc/2020/Conference — Accept (Poster)_

### Official Review · AnonReviewer3 · 2019-10-23
**Official Blind Review #3**

**Rating:** 3

**Review:**

# 1. Summary
The paper introduces a pre-training procedure for visual-linguistic representations. The model is an extension of BERT (with transformer backbone) to deal with visual input. Images are encoded using object detectors which regions are masked at pixel level. Experiments show state-of-the-art results on different downstream tasks.

Strengths of the paper:
* State-of-the-art results on 3 vision-language tasks

The weak reject decision was mainly guided by the following two weaknesses of the paper:
* Clarity of the paper needs to be improved to make the readers understanding the details of the model (see point 2 below)
* Limited novelty: the paper is an extension of BERT to the visual domain (see point 3 below)


# 2. Clarity
The paper reads quite well, although some points need to be improved:
* How were words split in sub-words (Sec 3.2)?
* "For each input element, its embedding feature is the summation of four types of embedding, ...": it is not clear how you sum embeddings. E.g., token embedding has 30k dimensions while image one has 2048 dimensions.
* "It is attached to each of the input elements, which is the output of a fully connected layer taking the concatenation of visual appearance feature and visual geometry embedding as input" -> this is not clear; what output are we talking about? What is the geometry embedding? I suggest to describe the two features first and then say at the end of the paragraph that the representation is the concatenation.
* "For the non-visual elements, the corresponding visual appearance features are of features extracted on the whole input image" -> what is the intuition of having the full image here? Some terms do not need to have an image associated (e.g., verbs or articles). Do you take care somehow of that?
* Once textual embeddings are masked by [MASK], the related visual embedding (whole image) is also masked? To my understanding the answer is no: what's the intuition of this?
* Segment embedding: is this important? This should be easy to show with an experiment in the ablation study of Table 4?
* It seems that there is a semantic asymmetry of input to the loss during training when considering only the text information (bookscorpus) and the image-text information (conceptual captions): how is training coping with this? Doesn't it make more sense to have 2 pre-training phases: first on text information only and then on image-text information?


# 3. Novelty and Motivation
The novelty of the paper is quite limited. It strongly relies on transformer networks and then recent success of BERT in the NLP domain. The proposal is an extension of these two ideas to visual domain.

Moreover, there is a body of concurrent work that is very similar to the proposed idea with slight differences (ViLBERT, VisualBERT, LXBERT, UNITER, B2T2), i.e., using transformers with masking operation on the RoIs. It is not clear what is the intuition related to the differences between the methods, i.e.
* Why one is better than the other; why should someone prefer this pre-training technique wrt others?
* Why a unified network (this work) is preferred wrt a two-stream one (ViLBERT, LXMERT)?
It seems that everything heavily depends on the experiments and empirical results obtained by trying many variants during the prototyping phase. It is missing a bit of understanding and intuition on the reasons why this technique should be used.


# 4. Experimentation
Experiments are the strength of the paper showing state-of-the-art results on 3 vision-language tasks. Some additional analysis is missing:
* If masking is conducted on the raw pixel, this makes training much slower since you need to perform inference many times. What is the impact in terms of accuracy? Did you carried out an experiment showing that it is better to mask raw pixels instead of conv maps?
* How long is the model trained for?
* What is the performance/accuracy on the pre-training tasks?
* How important is the segment embedding?
* Footnote 2 should be in the main text (Sec 4.1). It is too hidden, but very important to let the reader knowing about it.


**Experience Assessment:**

I have read many papers in this area.

**Review Assessment: Checking Correctness Of Derivations And Theory:**

N/A

**Review Assessment: Checking Correctness Of Experiments:**

I assessed the sensibility of the experiments.

**Review Assessment: Thoroughness In Paper Reading:**

N/A

---

> ### Author Response · Authors · 2019-11-14
> **Response to R#3**
>
> We feel we can well address the concerns of R#3, and hope R#3 give a second thought about the paper.
>
> Q#1: Concerns about novelty.
>
> A#1: First of all, the existence of concurrent works does not hurt the novelty of our method. And it should not be a reason for rejecting the paper. One cannot forecast what other research groups are doing when he/she conducts his/her own research.
>
> For better understanding of the readers, we even tried our best in comparing all the concurrent works in Related Works and in Appendix. The unique advantage of our work compared to other concurrent ones is presented at the end of Section 2. We quote the comments of R#1 here, “The paper does a decent job mentioning all the concurrent work in the space of learning multi-modal representations that have come out very recently.”
>
> In terms of comparison with BERT, we admit VL-BERT is an extension to the original BERT model. But it is non-trivial to extend BERT, designed for NLP tasks, to become a generic representation for visual-linguistic tasks.
>
> From the technology contribution perspective, numerous design choices are involved in VL-BERT for incorporating the visual information. It is interesting to note that in the previous work of VideoBERT, straight-forward design choices are made by directly turning video clips into visual words. The derived model is far from optimal. Our VL-BERT is well-designed to be: 1) a unified single-stream architecture, while also benefiting from single-modal pre-trained BERT and Fast R-CNN models; 2) end-to-end trainable with both the visual and linguistic branch parameters; 3) joint trained on both visual-linguistic and text-only corpus, so as to alleviate catastrophic forgetting [Kirkpatrick et. al., ``Overcoming catastrophic forgetting in neural networks.” PNAS, 2017.] of the text-only corpus in training networks for visual-linguistic tasks.
>
> From the practical importance perspective, we derived generic representations for various visual-linguistic tasks, which can be pre-trained on large-scale datasets. While previously various networks were designed specifically for different visual-linguistic tasks. The related discussions can be found in the Introduction and Related Works sections in the paper.
>
> Q#2: Concerns with clarity.
>
> A#2: In general, we feel many questions raised are because the topic is at the intersection of computer vision and NLP, with much preliminary knowledge involved. Such preliminary knowledge is self-explanatory in the corresponding domain but is unfamiliar for others.
>
> (a)	`` How were words split in sub-words?”
> The sub-words split is by WordPiece embeddings, which is a standard practice in NLP. For details, please refer to [Wu, Yonghui, et al. "Google's neural machine translation system: Bridging the gap between human and machine translation." arXiv preprint 2016.]
>
> (b)	Questions about token embedding.
> The token embedding is different from one-hot word embedding (e.g., 30k dim). It is of much lower dim (e.g., 768-d). The visual embedding is projected into the same dimension using a fully-connected layer as shown in Figure 1.
>
> (c)	To describe the two features first.
> Thanks for the suggestion. We would rearrange the paragraph in revision.
>
> (d)	What is the intuition of having the whole image embedding for textual input?
> To provide visual context for the sentence words.
>
> (e)	Once textual embeddings are masked by [MASK], the related whole image embedding is also masked?
> No, only textual embeddings are masked to block the linguistic input information.
>
> (f)	Is segment embedding important?
> Yes, as in BERT, the segment embedding is used to distinguish different input formats.
>
> (g)	How is training coping with the loss during training when considering text-only corpus and conceptual captions?
> There is no special treatment for the loss during training. The loss of Task #1 is averaged over the number of masked tokens. And the loss of Task #2 is averaged over the number of masked regions.
>
> (h)	Doesn't it make more sense to have 2 pre-training phases.
> Our experiments without Text-only Corpus is actually the ‘2 pre-training phases’ suggested by R#3. This is because our VL-BERT is initialized from a text-only pre-trained BERT. In our full version of VL-BERT, we train VL-BERT on visual-linguistic datasets together with text-only corpus. This is for alleviating catastrophic forgetting of the text-only corpus in training networks for visual-linguistic tasks.
>
> Q#3: Questions about experimentation.
>
> A#3: We address one question due to space limit.
>
> (a)	Masking on raw pixels.
> The masking would not slow down training. In a mini-batch, given an image, some RoIs are randomly sampled to be masked ones. The pixels lying in all the masked RoIs are set as zeros in the image at once. While the training loss drives the network to predict the labels of all the masked RoIs.
>
> As for masking conv maps, we observe obvious overfitting due to information leakage.

---

### Official Review · AnonReviewer1 · 2019-10-24
**Official Blind Review #1**

**Rating:** 6

**Review:**

### Summary:

This paper propose a new model for learning generic feature representations for visual-linguistic tasks by pretraining on large-scale vision and language datasets like Conceptual Captions and language-only datasets like BookCorpus and English Wikipedia. They demonstrate that the pre-training procedure can help improve performance on down-streaming tasks like visual question answering, visual commonsense reasoning.

Overall I liked the design choices made in the presentation. Although the paper doesn't provide insights around what the representations have learned and how they differ from representations learned / used by existing methods, they have provided substantial evidence to suggest that pre-training helps in a lot of downstream tasks.

Although it's hard to evaluate the paper without putting it in context with other concurrent works that have come out recently, I tried my best to evaluate the merits of the paper in isolation.

### Strengths:

- The paper explores an interesting direction of learning generic feature representations for visual-linguistic tasks for down-streaming tasks. Traditionally, people learn feature representations from scratch for each downstream task which might not always be possible if the training data is limited.
- The paper does a decent job mentioning all the concurrent work in the space of learning multi-modal representations that have come out very recently. They distinguish the proposed method from existing work and also compare the performance of the proposed approach with concurrent work on downstream tasks showing performance on-par or better than existing methods.
- I liked some of the design choices made in the paper. (1)  Instead of training a separate  transformer network for each type of input segments (question, ans, caption, image, etc). This makes the model easily extensible to other tasks as long as the correct segment embeddings are used to identify different input sources. (2) They also use a separate embedding for visual features instead of a common embedding  for both language tokens and visual tokens.
- Unlike the pre-training task in concurrent work, the model was pre-trained not just on multi-modal datasets like conceptual captions but also on text-only corpus like BookCorpus and English Wikipedia. The authors claim that this leads them to learn better representations for longer sentences which they found useful for VCR task.

### Weaknesses:

- The authors claim that attention mechanism in cross-modal transformer by concurrent approaches is restrictive but doesn't give substantial evidence that this is true. What are the limitations for cross-modal attention mechanisms compared to a single transformer model as described in this paper.
- On the contrary, by having a cross modal architecture, they can pre-train each modality separately on unaligned data. For instance, the text only transformer can be trained using large text corpora similar to BERT while the image only transformer can be trained on big datasets like OpenImages, ImageNet etc
- While the paper gives a lot of empirical evidence that pre-training helps, it would have been interesting to develop an understanding of what the model is actually learning and how are these representations better than learning representations from scratch for each task. For instance maybe the authors can visualize attention similar to [1].

### Other questions:

- When training on text-only datasets, what is the input on Visual Feature Embedding since there are no associated images. The authors mention that for non-visual elements, the features are extracted on the whole image. It's still unclear what the associated visual features are for text-only datasets.
- One of the pre-training tasks is masked ROI classification but it assigns a hard label to each ROI feature. It might be interesting to instead try learn the entire probability distribution (the output of a pre-trained classifier) by either minimizing the KL-divergence or by using softmax with soft-targets.
- While the model was learnt on text-only data, as mentioned in the above section, will the model help from image-only datasets such as large-scale classification datasets?
- While the models are tested on vision-and-language datasets, will these generic representations also be useful for unimodal tasks?

**Experience Assessment:**

I have published one or two papers in this area.

**Review Assessment: Checking Correctness Of Derivations And Theory:**

N/A

**Review Assessment: Checking Correctness Of Experiments:**

I carefully checked the experiments.

**Review Assessment: Thoroughness In Paper Reading:**

I read the paper thoroughly.

---

> ### Author Response · Authors · 2019-11-14
> **Response to R#1**
>
> We thank the reviewer for the careful reviews and constructive suggestions. We address the questions as follows.
>
> Q#1: ``The authors claim that attention mechanism in cross-modal transformer by concurrent approaches is restrictive but doesn't give substantial evidence that this is true. What are the limitations for cross-modal attention mechanisms compared to a single transformer model as described in this paper.”
>
> A#1: There should be some misunderstanding here. The related discussion is on page 3. The description of ``restricted” is about the attention pattern in ViLBERT and LXMERT, not about their actual application scenarios or experimental results. Actually, the authors of ViLBERT deliberately designed such restricted attention modules. And they claimed such restricted attention is superior than a single-stream unified model.
>
> Meanwhile, we found our unified architecture based on Transformers without any restriction on the attention patterns achieves accuracy even superior than those in ViLBERT. More careful comparison would be made in the future.
>
> Q#2: ``On the contrary, by having a cross modal architecture, they can pre-train each modality separately on unaligned data. For instance, the text only transformer can be trained using large text corpora similar to BERT while the image only transformer can be trained on big datasets like OpenImages, ImageNet etc”
>
> A#2: Actually, our VL-BERT also benefits from pre-training on single modality pre-training. The original BERT parameters are pre-trained on large text-only corpus. While the Fast R-CNN parameters are pre-trained for image object detection. The related description is made in Section 4.1.
>
> Q#3: ``it would have been interesting to develop an understanding of what the model is actually learning and how are these representations better than learning representations from scratch for each task”
>
> A#3: Thanks for the suggestion. Visualization of attention map has been added in revision (See Appendix A.3), which shows the pre-training of VL-BERT learns the detailed alignment between visual and linguistic contents.
>
> Q#4: `` It's still unclear what the associated visual features are for text-only datasets.”
>
> A#4: The visual feature input for textual corpus is a learnable embedding shared for all words. We shall clarify in revision.
>
> Q#5: `` One of the pre-training tasks is masked ROI classification but it assigns a hard label to each ROI feature. It might be interesting to instead try learn the entire probability distribution”
>
> A#5: Thanks for the suggestion, we shall try.
>
> Q#6: Will the model help from image-only datasets? Will the generic representations also be useful for unimodal tasks?
>
> A#6: Thanks for the great suggestions. Actually, we are currently working hard towards this direction.

---

### Official Review · AnonReviewer2 · 2019-10-26
**Official Blind Review #2**

**Rating:** 6

**Review:**

This paper proposed a pre-trainable generic representation for visual-linguistic tasks call VL-BERT. VL-BERT extend BERT by changing the input from subsequent sentence to image regions and modify the caption words has the additional visual feature embedding. The authors pre-train the VL-BERT on the conceptual caption dataset and Wikipedia and book corpus dataset, empirical results show that the VL-BERT achieve the SOTA performance on the VCR, VQA and refer expression tasks.

As the authors mentioned in Table 5, pre-training the visolinguistic representation for vision and language tasks is very popular recently, and 5~6 similar works have appeared recently. One of the nice features I found on this work is it's joint train with text-only corpus and faster RCNN weight. While ViLBERT designs for easier extendable for other modalities, VLBERT is more focus on the representation learning on the vision and language, since the caption input also combines with the visual feature embedding.

Overall the paper is well written and performs extensive experiments/ablations. There is some specific point that is not clear to me or needs further clarifications from the authors.

1: The authors mentioned the improvement over tuning the visual parameters, I wonder what is the details on that? is the region proposal network's weight fixed? if not, how to avoid the shift on the proposal layer? Is the model still has the visual genome target or objective? Which layer is fixed/updated? and what is the optimizer and learning rate scheduler?

2: I notice there is a change in the textual input which take visual feature embeddings. I wonder what is the performance without these features? What is the visual feature input for textual corpus?

3: For the Masked RoI classification with Linguistic Clues, what if there are overlapped regions? what if the detection label from faster rcnn is incorrect? will this introduce any noise?

4: For VCR tasks, it seems the VL-BERT_base w/o pre-training is performed similar compare to the with pre-training (only 0.7% lower on val of Q->A) I wonder what is the reason of this? Is this show the pre-training is not important for the VCR tasks?

5: The VCR tasks also have the object bounding box correspondence, is VL-BERT take any of this supervision for input? If not, how does the VL-BERT learn the correspondence?

6: For refer expression tasks, the VL-BERT_base is actually worse than ViLBERT on the detected regions. It's not a fair comparison since other models use bert-base model.

Overall, I think this paper is well written and has solid experiment results. It will be great if the authors can further clarify the above questions.

**Experience Assessment:**

I have published one or two papers in this area.

**Review Assessment: Checking Correctness Of Derivations And Theory:**

N/A

**Review Assessment: Checking Correctness Of Experiments:**

I assessed the sensibility of the experiments.

**Review Assessment: Thoroughness In Paper Reading:**

I read the paper at least twice and used my best judgement in assessing the paper.

---

> ### Author Response · Authors · 2019-11-14
> **Response to R#2**
>
> We thank the reviewer for the careful reviews and constructive suggestions. We clarify the questions as follows.
>
> Q#1: Details on tuning the visual parameters.
>
> A#1: As described in Section 3.2 and Fig. 1, in VL-BERT, only the object detection branch (i.e., Fast R-CNN) in Faster R-CNN is exploited to extract visual feature embeddings for each RoI. The region proposal network is not involved in training/inference of VL-BERT. The optimizer and learning rate mentioned in experiments are shared for all the parameters in VL-BERT and Fast R-CNN.
>
> Indeed, such a training scheme would cause shift on the proposal layer in Faster R-CNN. Here we extract the RoIs on the training/test samples by a separate pre-trained Faster R-CNN. The shift may well be alleviated by joint training on object detection tasks.
>
> Q#2: ``What is the visual feature input for textual corpus?” ``I wonder what is the performance without these features?”
>
> A#2: The visual feature input for textual corpus is a learnable embedding shared for all words. We did not try experimenting without such features. We shall clarify in revision.
>
> Q#3: What if there are overlapped regions in task Masked RoI classification with Linguistic Clues? What if the detection label from the pre-trained Faster R-CNN is incorrect?
>
> A#3: The pixels laid in the masked RoI are set as zeros, regardless of whether the pixels also belong to other RoIs. Thus, for the pixels covered by overlapping RoIs, they are also simply set as zeros. The detection labels on Conceptual Captions can be incorrect, since they are just pseudo labels generated by a pre-trained Faster R-CNN. Because there are no ground-truth detection annotations on Conceptual Captions, we cannot validate the effect for the time being.
>
> Q#4: Is the pre-training not important for the VCR tasks?
>
> A#4: We believe this is because the pre-training task on Conceptual Captions is for image captioning, where no commonsense reasoning is involved, which is vital for the VCR task.
>
> Q#5: Does VL-BERT use object bounding box correspondence annotations for VCR dataset?
>
> A#5: No, we did not use the bounding box correspondence annotations for VCR dataset, for the coherence in the design of VL-BERT. We also tried exploiting the annotated bounding box correspondence in VCR, but there is little difference in accuracy. We feel VL-BERT might already learned to encode such correspondence.
>
> Q#6: VL-BERT is actually worse than ViLBERT on refer expression tasks
>
> A#6: Yes, VL-BERT is slightly shy of ViLBERT on RefCOCO+. Meanwhile, VL-BERT and ViLBERT are concurrent works. We feel there is no problem that VL-BERT does not surpass ViLBERT on every benchmark.

---

### Author Response · Authors · 2019-11-14
**A new section Appendix A.3 for visualization of attention maps has been added**

A new section Appendix A.3 for visualization of attention maps has been added.

---

### Decision · Program_Chairs · 2019-12-19

**Decision:**

Accept (Poster)

**Comment:**

The paper proposed a new pretrained language model which can take visual information into the embeddings. Experiments showed state-of-the-art results on three downstream tasks. The paper is well written and detailed comparisons with related work are given. There are some concerns about the clarity and novelty raised by the reviewers which is answered in details and I think the paper is acceptable.